# Prevalence and Duration of Use of Medicines Recommended for Short-Term Use in Aged Care Facility Residents

**DOI:** 10.3390/pharmacy7020055

**Published:** 2019-06-06

**Authors:** Lisa M. Kalisch Ellett, Gizat M. Kassie, Nicole L. Pratt, Mhairi Kerr, Elizabeth E. Roughead

**Affiliations:** Quality Use of Medicines and Pharmacy Research Centre, School of Pharmacy and Medical Sciences, University of South Australia, GPO Box 2471, Adelaide, SA 5001, Australia; lisa.kalisch@unisa.edu.au (L.M.K.E.); Nicole.Pratt@unisa.edu.au (N.L.P.); Mhairi.Kerr@unisa.edu.au (M.K.); Libby.Roughead@unisa.edu.au (E.E.R.)

**Keywords:** residential aged care facility, nursing home, inappropriate medication use, appropriate medication use, medication review, potentially inappropriate medicines, over prescribing, older people

## Abstract

**Background**: Multiple studies have assessed the appropriateness of the use of medicines for nursing home residents; however, few have included duration of use in their assessment. The aim of this study was to assess the level and duration of use of medications recommended for short-term use in residents of aged care facilities in Australia. **Methods**: Australian Government Department of Veterans’ Affairs (DVA) administrative claims data were used for this study. Veterans eligible for all health services subsidised by DVA were followed for one year from 1 July 2015 to 30 June 2016. The number of days covered for each medicine was calculated by multiplying the number of prescriptions dispensed during the year by the pack duration for the medicine. The pack duration was calculated by dividing the quantity supplied at each dispensing by the usual number of doses per day in older people according to Australian prescribing guidelines. The proportion of patients using each medicine and the number of days covered during the study period were determined. **Results**: 14, 237 residents met the inclusion criteria. One in five participants were dispensed antipsychotics, and the median duration of use was 180 days in the one-year period. More than one-third were dispensed a benzodiazepine, and the median duration of use was 240 days in the year. Half were dispensed an opioid analgesic with a median duration of use of 225 days in the year. Fifty-two percent were dispensed proton pump inhibitors with a median duration of use of 360 days in the year. A quarter received an antibiotic recommended for the management of urinary tract infection, with a median duration of use of 14 days in the year. **Conclusion**: Long-term use of antipsychotics, benzodiazepines, opioid analgesics and proton pump inhibitors is common in aged care residents. Ensuring appropriate duration of use for these medicines is necessary to reduce risk of harm.

## 1. Introduction 

Older people frequently have multiple comorbidities, and in turn, they commonly use both short- and long-term medications to manage these conditions [1]. In one nation-wide cross-sectional survey involving 1600 Australians aged 50 years or older, 43% reported that they had used five or more medicines in the past 24 hours [2]. As the number of medications being concurrently taken by a patient increases, the likelihood of inappropriate use also increases [3,4]. A study involving 4373 patients in nursing homes in Norway found that the number of medicines used was significantly associated with the likelihood of using potentially inappropriate medicines (r = 0.43, p < 0.001) [5]. Prior research conducted in Canada has also shown that aged care facility residents tend to use multiple medicines [6]. A systematic review which included 44 studies from 22 countries found that up to 91% of long-term facility residents use five or more medicines and 65% take eleven or more medicines concurrently [7]. 

A number of tools have been developed to assess the appropriateness of medicine use by older people; including the Beers criteria [8] and Screening Tool of Older People’s Prescriptions (STOPP)/ Screening Tool to Alert to Right Treatment (START) criteria [9]. A systematic review which included 48 studies, conducted between 1990 and 2015, reported that the overall prevalence of use of potentially inappropriate medicines was 43% in nursing home residents [10]. In contrast, two systematic reviews that assessed studies on potentially inappropriate use of medicines by community-dwelling older adults reported a lower prevalence of approximately 20% [11,12], indicating that use of potentially inappropriate medicines is more common in nursing home residents than in those living in the community. 

The use of potentially inappropriate medicines has been associated with adverse clinical outcomes including mortality, falls, urinary tract infections, drug-drug interactions, and increased cost [13,14,15]. The risk of harm associated with the use of many of these medicines, including proton pump inhibitors (PPIs), benzodiazepines, antipsychotics and non-steroidal anti-inflammatory drugs (NSAIDs), increases with prolonged use [4,16]; however, when used short term for specific indications, the benefits to use may outweigh these risks for some patients. For example, the 2019 updated Beer’s Criteria indicate that, if used at all, proton pump inhibitors and NSAIDs should be used for short periods of time in older patients [8], and the STOPP/START criteria recommend that benzodiazepines, NSAIDs for osteoarthritis and PPIs should be used for a duration of less than four weeks, three months and two months, respectively [9]. However, people commonly use these medicines for longer than recommended. A retrospective evaluation of prescription claims data for 338,801 older Irish people, living both in the community and in aged care facilities, found that 17% of patients used proton pump inhibitors for more than two months, 9% used NSAIDs for more than three months and 5% used benzodiazepines for more than one month [17]. 

Although there is evidence for harm associated with long-term use of these medicines, no studies have assessed the duration of use of medicines recommended for short-term use in Australian residents in aged care facilities. Therefore, the aim of this study was to assess the duration and prevalence of use of medications recommended for short-term use in residents of aged care facilities in Australia, with a focus on antipsychotics, benzodiazepines, opioid analgesics, antiemetics, NSAIDs, PPIs and antibiotics for urinary tract infections.

## 2. Methods

Veterans who were eligible for all health services subsidised by the Australian Government Department of Veterans’ Affairs (DVA) (‘gold card’ holders) and residing in continuous aged care at the study start date were included. People receiving respite care, rehabilitation or in the community and those who died prior to the end of the study period were excluded from the analyses. The study period was from 1 July 2015 to 30 June 2016. Ethics approval was obtained from the University of South Australia Human Research Ethics Committee and the Departments of Defence and Veterans’ Affairs Human Research Ethics Committee.

We reviewed Australian prescribing references, including the Australian Medicines Handbook [18], and potentially inappropriate medicines lists, including the STOPP/START criteria [9] and the Beers criteria [8], to identify medicines which are recommended for short-term use in older people (Table 1). 

The Australian Government Department of Veterans’ Affairs (DVA) healthcare claims data were used for this study. The DVA administrative claims database contains details of all prescription medicines, medical, allied health services and hospitalisations provided to eligible Australian war veterans, their spouses and dependents. In 2016, the data file contained records for a treatment population of approximately 250,000 members of the veteran community. DVA maintain a client file, which includes data on gender, date of birth, date of death and family status. Medicines are coded in the dataset according to the World Health Organization (WHO) Anatomical Therapeutic Chemical (ATC) classification [19] and the Schedule of Pharmaceutical Benefits item codes [20]. Hospitalisations are coded according to the WHO International Classification of Diseases-10th Edition, Australian Modification [21]. 

The percentage of participants dispensed the medications of interest and the median days covered during the study follow-up period were calculated. Prescription duration is not recorded in the DVA data set, so the number of days covered for each study medicine (Table 1) was calculated by multiplying the number of prescriptions dispensed during the year by the pack duration for the respective medicine. Aged care facility residents in Australia usually receive their medicines packed in dose administration aids. When the resident runs out of a medicine, pharmacists are able to re-dispense using a repeat prescription or an ‘owing’ prescription if no repeats are available to ensure continuity of supply [22]. Any gaps in continuity of supply of medicines to aged care facility residents are likely to represent true breaks or stops in medicine use, rather than poor compliance or delayed supply. Therefore, pack duration was calculated by dividing the quantity supplied at each dispensing by the usual number of doses per day in older people, from Australian prescribing guidelines [23]. The number of days covered was a maximum of 365 days, that is, coverage for the entire study period. 

The demographics and clinical data of participants including gender and median age at study entry were reported and the median number of co-morbidities was determined based on the medicines dispensed at study entry using a validated instrument, RxRisk-V8 [24]. Participants’ general practitioner and specialist visits, number of prescriptions and unique medicines dispensed during the study period were also reported. All analyses were performed using SAS for windows, V9.3 SP4 (SAS institute, Cary, North Carolina, USA). 

## 3. Results 

There were 14,237 aged care residents who met the inclusion criteria for this study. The median age at study entry was 91 years (interquartile range (IQR), 88–93) and 10,833 (76.1%) of the participants were women. Study participants had a median of 77 prescriptions (IQR, 48–111) dispensed for a median of 12 unique medicines (IQR, 8–17) during the one-year study period, and they had a median of five comorbidities (IQR, 4–7).

One in five aged care facility residents were dispensed an antipsychotic during follow-up, most commonly risperidone (12% of participants). The median number of days covered for risperidone was 240 days, and a quarter of residents who were dispensed risperidone were covered for the entire year of follow-up. Although olanzapine was dispensed to only 3% of residents, half of them were dispensed enough olanzapine to last for more than 336 days and a quarter were covered for the entire one-year study period. Three hundred and fifty one residents (2%) were dispensed haloperidol and the median number of days covered was 100. More than one-third of the residents were dispensed a benzodiazepine during the one-year period and the median number of days covered was 240 (Table 2). A quarter of those dispensed a benzodiazepine received enough medicine for continuous use over the entire year. 

At least half of the residents were dispensed analgesics (50% opioid analgesics and 8% NSAIDs). The most commonly used opioid analgesic was oxycodone (28% of residents) with a median duration of use of approximately one month. Buprenorphine patches were also commonly used (19% of participants) and one-quarter of residents dispensed buprenorphine patches received enough medicine for continuous use over the entire one-year study period. A small proportion of patients were dispensed antiemetics. The median duration of use of metoclopramide (dispensed to 17% of residents) and prochlorperazine (dispensed to 5% of residents) was 8 days. In contrast, the median duration of use of domperidone (dispensed to 4% of residents) was longer, at just over two-months. Over half of the study cohort (52%) were dispensed proton pump inhibitors (PPIs), and half of these people received enough medicine for continuous use over the entire one-year study period (Table 2).

One quarter of the residents received an antibiotic recommended for the treatment of urinary tract infections during the study period, most commonly trimethoprim (dispensed to 20% of residents) with a median duration of use of seven days (Table 2). 

## 4. Discussion 

A number of medicines have recommendations for short-term duration of use in older people. Our study has shown that for some of these medicines, over the course of a year, many aged care residents use these medicines for longer than the recommended duration. Of note, one quarter of aged care residents in our study who were dispensed risperidone, benzodiazepines or proton pump inhibitors received enough medicine for continuous use for the entire one-year study period. This long-term use occurred despite prescribing guidelines indicating that use of these medicines should not exceed 12 weeks (risperidone), 4 weeks (benzodiazepines) or 8 weeks (PPIs). The benefits associated with long-term use of these medicines are unlikely to outweigh the risks. For example, the risk of hip fracture increases with long-term use of PPIs [16] and the risk of developing delirium is associated with prolonged use of benzodiazepines [8,26]. These adverse events are associated with impaired functional and cognitive status, which contribute to lower quality of life for older patients. 

One in five residents in the aged care facilities in our study (20%) were dispensed an antipsychotic during the one-year study period. Risperidone was the most commonly dispensed antipsychotic (12% of participants), and this likely reflects Australian subsidy restrictions for antipsychotics. Risperidone is the only subsidised antipsychotic for behavioural disturbances in dementia in Australia [27]. Dementia is the most likely indication for antipsychotic use in the aged care population, with approximately half of aged care residents in Australia diagnosed with dementia [28]. Since June 2015, the use of risperidone in dementia has been restricted to a maximum of 12 weeks and only for people with Alzheimer’s dementia [18], due to the increased risk of stroke associated with longer term use and in other types of dementia [29]. Our analysis covered the one-year time period immediately after this change in the recommended duration of use, and the median duration of risperidone use in study participants of our study was 29 weeks, more than double the recommended duration of use. 

The median duration of benzodiazepine use in our study population, which was approximately 8 months, also exceeded the recommended one month duration of use [18]. The use of benzodiazepines in older people is associated with increased risk of adverse events, including falls [30] and delirium [31]. Older patients are already at high risk of falls, fracture and delirium [32,33]; and prolonged exposure to benzodiazepines further increases the risk of these adverse events [8]. The prevalence of benzodiazepine use in our study (34%) was similar to the prevalence range of use reported in other international (Belgium, 54%) [1] and Australian (21%) [34] studies conducted in residents of aged care facilities. 

Over half of the participants in our study were dispensed proton pump inhibitors (PPIs) and the median duration of use was nearly one year (median 360 days). Evidence shows that the use of proton pump inhibitors is significantly associated with increased risk of hip fracture (OR = 2.65; 95% CI = 1.80–3.90; P < 0.001) and the risk increases with prolonged use [16]. Prolonged use of proton pump inhibitors was also found to be associated with kidney disease [35] and intestinal bacterial infections [36]. The prevalence of PPI use in this study is higher than the prevalence reported in previous studies conducted in nursing home residents in the United States (27%) [37] and France (38%) [38]. Previous nation-wide intervention programs in Australia that aimed to reduce the duration of use and dose of PPIs in older patients have resulted in significant changes in the clinical practice [39,40], so the level and duration of PPI use could have been higher than what was observed in this study if these interventions had not been implemented. The extended use of proton pump inhibitors (median duration of use = 360 days) observed in this study shows that the use of these medicines in residents of aged care facilities in Australia requires clinicians’ follow-up and regular review. 

Although risperidone, benzodiazepines and PPIs were commonly used long-term by participants in our study, the median duration of use of many of the other medicines aligned with the prescribing recommendations. For example, one quarter of study participants received an antibiotic for UTI during the study period and in most cases the median duration of use in the year was under two weeks. Nitrofurantoin was the only antibiotic with a longer median duration of use (median 30 days) and this antibiotic is indicated for UTI prophylaxis; so long term use may be appropriate for some patients, particularly those requiring prophylaxis for recurrent UTIs [18]. 

In our study, we also found that in most cases antiemetics were used short-term. Although metoclopramide use was common (17% of participants), the median duration of use was only 8 days. The median duration of use of domperidone was higher, at 69 days. If used for nausea and vomiting the recommended duration of use of domperidone is seven days; however, if used for the management of gastroparesis, up to four-weeks use is indicated [18]. Diagnoses are not recorded in our dataset, so we cannot determine whether the longer duration of use of domperidone was appropriate (for gastroparesis) or potentially inappropriate (for nausea and vomiting); although the overall prevalence of domperidone use in our study is much higher than the estimated population prevalence of gastroparesis [41]. The risks associated with long-term use of domperidone include depression, anxiety, somnolence, headache, akathisia, and diarrhoea [42]. The use of domperidone in older patients has also been found to be associated with a significantly increased risk of cardiac arrhythmia and sudden cardiac death [42,43]. Therefore, the benefits associated with long-term use may not outweigh the risks. 

There are some limitations that need to be considered when interpreting the findings of this study. We could not evaluate the appropriateness of use of medications in terms of indication and dose, as these data are not available in the administrative claims database used for this study. However, the findings in this study are informative in that it included the number of days covered for each medication within the one-year follow-up period, data which is frequently missing in medication utilisation studies in nursing home residents. The study included all medicines dispensed over the one-year study period, which avoided seasonal variation in use of medicines and included a large sample size, which improves the generalisability of the results.

## 5. Conclusions

Although guidelines recommend short term use of benzodiazepines, risperidone, opioids and proton pump inhibitors amongst older people, if they are used at all, results of our study show that a large number of aged care residents use these medicines and that prolonged use of these medicines is common. Long-term use of these medicines may lead to increased risk of adverse events. Ensuring appropriate duration of use of these medicines is necessary to reduce risk of harm associated with longer-term use. Efforts to improve the use of medicines recommended for short-term need to be instituted in Australian aged care facilities, including education programs, regular review of medications by pharmacists and the use of guidelines supporting appropriate medication use in older patients, such as the Beers and START/STOPP criteria. 

## Figures and Tables

**Table 1 pharmacy-07-00055-t001:** Prescribing recommendations for study medicines.

Medication	Australian Prescribing Recommendations [18,25]	Beers Criteria Recommendations [8]	STOPP/START Criteria Recommendations [9]
**Antipsychotics**	When used for the management of the behavioural and psychological symptoms of dementia (BPSD), risperidone should be used for no longer than 12 weeks	Avoid, except for schizophrenia, bipolar disorder, or short-term use as antiemetic during chemotherapy	Avoid use of neuroleptic antipsychotic in patients with behavioural and psychological symptoms of dementia (BPSD) unless symptoms are severe and other treatments have failed (increased risk of stroke) Avoid use of antipsychotics (i.e., other than quetiapine or clozapine) in those with parkinsonism or Lewy Body Disease (risk of severe extra-pyramidal symptoms)
**Benzodiazepines**	Benzodiazepines should be used short term, i.e., for 2–4 weeks duration or intermittent use only	Avoid use in older people	Avoid use of benzodiazepines for ≥ 4 weeks (risk of prolonged sedation, confusion, impaired balance, falls, road traffic accidents)
**Analgesics**	Non-steroidal anti-inflammatory drugs (NSAIDs): Avoid all NSAIDs if there is a history of gastrointestinal bleeding, or use with extreme caution and use prophylaxis Opioids: For chronic, non-cancer pain: start as a 4–8-week trial and taper if no benefit	NSAIDs: Avoid chronic use, unless other alternatives are not effective and patient can take gastroprotective agent (proton-pump inhibitor or misoprostol) Avoid indomethacin use as it has the most adverse effects of all NSAIDS Opioids: Avoid (in people with history of falls or fractures), excludes pain management due to recent fractures or joint replacement	Avoid use of non-cyclooxygenase-2 selective non-steroidal anti-inflammatory drug with history of peptic ulcer disease or gastrointestinal bleeding, unless with concurrent proton pump inhibitors (PPI) or H2 antagonist (risk of peptic ulcer relapse) Avoid using NSAIDs if estimated glomerular filtration rate < 50 mL/min/1.73 m^2^ (risk of deterioration in renal function) Avoid use of oral or transdermal strong opioids (morphine, oxycodone, fentanyl, buprenorphine, diamorphine, methadone, tramadol, pethidine, pentazocine) as first line therapy for mild pain (WHO analgesic ladder not observed) Avoid use of regular (as distinct from ‘*as required*’) opioids without concomitant laxative (risk of severe constipation) Avoid use long-acting opioids without short-acting opioids for break-through pain (risk of non-control of severe pain)
**Antiemetics**	Metoclopramide: Maximum length of treatment is 5 days Domperidone: For nausea and vomiting, treatment duration should not exceed one week Prochlorperazine: Use short-term only, to reduce risk of tardive dyskinesia	Metoclopramide: avoid, unless for gastroparesis with duration of use not to exceed 12 weeks except in rare cases Prochlorperazine: avoid, except for short-term use as antiemetic during chemotherapy	Avoid use of prochlorperazine or metoclopramide with Parkinsonism (risk of exacerbating Parkinsonian symptoms)
**PPIs**	When used for maintenance therapy in dyspepsia or gastro-oesophageal reflux disease the need for ongoing treatment should be regularly reviewed, with the aim to cease use altogether or use intermittently or at a reduced dose if symptoms are well controlled	Avoid scheduled use for more than 8 weeks unless for high-risk patients (e.g., oral corticosteroids or chronic NSAID use), erosive esophagitis, Barrett esophagitis, pathological hypersecretory condition, or demonstrated need for maintenance treatment (eg, because of failure of drug discontinuation trial or H2-receptor antagonists)	Avoid use of PPIs for uncomplicated peptic ulcer disease or erosive peptic oesophagitis at full therapeutic dosage for > 8 weeks (dose reduction or earlier discontinuation indicated)
**Antibiotics for urinary tract infections (UTI)**	Trimethoprim: prophylaxis for urinary tract infections can be continued for 3 to 6 months, or in some cases for longer periods. Nitrofurantoin: Long-term treatment needs monitoring for pulmonary and liver function (every month for 3 months, then every 3 months) Ciprofloxacin: prophylaxis for urinary tract infections can be continued for 3 to 6 months, or in some cases for longer periods.	Nitrofurantoin: avoid use for long-term suppression of bacteria or in individuals with creatinine clearance < 30 mL/min Ciprofloxacin: avoid or reduce dose if creatinine clearance < 30 mL/min due to increased risk of central nervous system effects (e.g., seizure, confusion) and tendon rupture Trimethoprim-sulfamethoxazole: avoid if creatinine clearance < 15 mL/min and reduce dose if creatinine clearance < 30 mL/min due to increased risk of worsening of renal function and hyperkalemia	

**Table 2 pharmacy-07-00055-t002:** Use of study medicines by aged care facility residents (n = 14,237).

Medicine (ATC Code)	Number (%) Residents Dispensed Medicine	Median (p25, p75) Days Covered
**Any antipsychotic (N05A, excluding N05AB04)** *	2873 (20%)	180 (75, 300)
Risperidone (N05AX08)	1726 (12%)	240 (120, 365)
Quetiapine (N05AH04)	531 (4%)	180 (60, 270)
Olanzapine (N05AH03)	499 (3%)	336 (168, 365)
Haloperidol (N05AD01)	351 (2%)	100 (50, 300)
**Benzodiazepines (N05BA, N05CD)** *	**4821 (34%)**	**240 (60, 365)**
Temazepam (N05CD07)	2953 (21%)	175 (50, 350)
Oxazepam (N05BA04)	1709 (12%)	150 (50, 325)
Diazepam (N05BA01)	723 (5%)	33 (17, 100)
Nitrazepam (N05CD02)	186 (1%)	250 (100, 350)
Alprazolam (N05BA12)	128 (1%)	100 (67, 183)
**Analgesics**		
**Any opioid analgesic (N02A)** *	**7049 (50%)**	**225 (45, 365)**
Buprenorphine patches (N02AE01)	2676 (19%)	350 (126, 365)
Oxycodone (N02AA05)	4031 (28%)	33 (5, 224)
Fentanyl patches (N02AB03)	910 (6%)	364 (197, 365)
Tramadol (N02AX02)	714 (5%)	23 (5, 140)
Morphine (N02AA01)	113 (1%)	182 (70, 364)
**Any non-steroidal anti-inflammatory drug (M01A)** *	1114 (8%)	45 (15, 135)
Meloxicam (M01AC06)	454 (3%)	90 (30,270)
Celecoxib (M01AH01)	304 (2%)	60 (30,180)
Ibuprofen (M01AE01)	149 (1%)	30 (10, 30)
Diclofenac (M01AB05)	79 (0.6%)	50 (25, 100)
Naproxen (M01AE02)	39 (0.3%)	25 (14,50)
Indomethacin (M01AB01)	29 (0.2%)	50 (50, 50)
Piroxicam (M01AC01)	8 (0.06%)	50 (38,150)
**Antiemetics**		
Metoclopramide (A03FA01)	2464 (17%)	8 (3, 17)
Prochlorperazine (N05AB04)	737 (5%)	8 (8, 33)
Domperidone (A03FA03)	554 (4%)	69 (19, 175)
**Proton pump inhibitors (A02BC)**	7383 (52%)	360 (330, 365)
Pantoprazole (A02BC02)	3150 (22%)	360 (240, 365)
Esomeprazole (A02BC05)	2634 (19%)	360 (300, 365)
Omeprazole (A02BC01)	1028 (7%)	360 (270, 365)
Rabeprazole (A02BC04)	798 (6%)	360 (270, 365)
Lansoprazole (A02BC03)	241 (2%)	336 (168, 364)
**Antibiotics for UTI** **	3382 (24%)	14 (7,30)
Trimethoprim (J01EA01)	2834 (20%)	7 (7, 14)
Norfloxacin (J01MA06)	494 (4%)	7 (7, 14)
Nitrofurantoin (J01XE01)	634 (4%)	30 (30, 90)

* Results presented for most frequently dispensed medicines in this class; ** Antibiotics for UTI were trimethoprim, norfloxacin or nitrofurantoin as these medicines are only recommended for use in UTI. Although other antibiotics like ciprofloxacin or cephalexin are recommended for UTI, they are also used in the management of other infections and so are not presented here.

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
