# Peer review of "Prevalence and Duration of Use of Medicines Recommended for Short-Term Use in Aged Care Facility Residents"

_pharmacy, 2019, doi:10.3390/pharmacy7020055_

Reviewer 1 Report

This is a survey of medication usage in long-term care facilities among Australian Department of Veterans Affairs patients.

The authors provide important information suggesting that patients continue to be prescribed potentially inappropriate medications despite the propagation of recommendations for best practice.

Please detail the population being studied. For instance are these long-term care patients, including dementia care?   Are rehabilitation patients included? This helps the reader understand added implications of the prescribed medications.

In the discussion,the 2 paragraphs between lines 162 and 181 should be inserted in line  214 as the points raised should follow discussion of your major findings.

Can you amplify your conclusion? Perhaps a stronger statement related to prescriber education, the need for pharmacy review of long term care patients, stress the need for patient safety and applicability of Beers and START/STOPP criteria to Austrailian patients, or perhaps some policy recommendations?

Author Response

Dear editor,                                               

We are thankful for the reviewer’s constructive feedback on our manuscript. We found the comments insightful and we have revised the manuscript accordingly. We have shown the amendments in track changes on the manuscript attached on the site. Our point by point response to the comments/suggestions from reviewer 1 is attached here.

Reviewer 2 Report

Thank you for submitting your manuscript “Prevalence and duration of use of medicines recommended for short-term use in aged care facility residents”. Polypharmacy is common among older people and can reduce their quality of life.

In the abstract, the authors state how they will calculate medicine duration. “The number of days covered for each medicine was calculated by multiplying the number of prescriptions dispensed during the year by the pack duration for the medicine.” This suggests that only one tablet a day was taken? Possibly the definition from the methods is clearer for the abstract “pack duration was calculated by dividing the quantity supplied at each dispensing by the usual number of doses per day in older people ……” 

In the introduction, regarding the statement “Prior research has also shown that aged care facility residents tend to use multiple medicines (6, 7).” were these studies conducted in Australia? Otherwise, they would duplicate this research?

In the methods, I suggest grouping similar contents together e.g. the first paragraph includes data coding which traditionally would be described later in the methods. I also recommend noting that the prescription guidelines were the ones related to the data period e.g. chronic pain guidelines have been updated and now discourage opioid use.

Results

How many people were in the database? Was there much change in the number of people in the database during the sampling period? For example, did more people die than join?

What was the number of people excluded not fitting the selection criteria? 

Results are described by first stating the type of medicine which is very helpful or the reader. This is not consistent in the third paragraph, which includes opioids and metoclopramide etc. I suggest writing the text using similar categories as in Table 2 e.g. analgesics etc.

Regards the data in Table 2, are they the top medicines prescribed? The footnotes suggest it is, but it is not ordered that way. How is the data ordered in the table beyond medicine groups? 

Does the data detail what medicines were co-prescribed together? For instance, the use of opioid analgesics and benzodiazepines should be avoided. The discussion mentions associated falls risks with the prescription of benzodiazepines.

Author Response

Dear editor,                                               

We are thankful for the reviewer’s constructive feedback on our manuscript. We found the comments insightful and we have revised the manuscript accordingly. We have shown the amendments in track changes on the manuscript attached on the site. Our point by point responses to the comments/suggestions from reviewer 2 is attached here.
